# Climate change may induce connectivity loss and mountaintop extinction in Central American forests

Lukas Baumbach [1] ✉, Dan L. Warren [2], Rasoul Yousefpour[1] & Marc Hanewinkel [1]

The tropical forests of Central America serve a pivotal role as biodiversity hotspots and provide ecosystem services securing human livelihood. However, climate change is expected to affect the species composition of forest ecosystems, lead to forest type transitions and trigger irrecoverable losses of habitat and biodiversity. Here, we investigate potential impacts of climate change on the environmental suitability of main plant functional types (PFTs) across Central America. Using a large database of occurrence records and physiological data, we classify tree species into trait-based groups and project their suitability under three representative concentration pathways (RCPs 2.6, 4.5 and 8.5) with an ensemble of state-of-the-art correlative modelling methods. Our results forecast transitions from wet towards generalist or dry forest PFTs for large parts of the study region. Moreover, suitable area for wet-adapted PFTs is projected to latitudinally diverge and lose connectivity, while expected upslope shifts of montane species point to high risks of mountaintop extinction. These findings underline the urgent need to safeguard the connectivity of habitats through biological corridors and extend protected areas in the identified transition hotspots.

[1] Chair of Forestry Economics and Forest Planning, University of Freiburg, Freiburg, Germany. [2] Biodiversity and Biocomplexity Unit, Okinawa Institute of Science and Technology, Onna-son, Okinawa, Japan. ✉email: lukas.baumbach@ife.uni-freiburg.de

Evidence of climate change impacts on vegetation growth and distribution has been accumulating all around the globe. At the current pace of environmental changes, many species may be unable to adapt to the new conditions, eventually leading to habitat range shifts or their extinction[1–3]. Beyond ecological consequences, such shifts may significantly alter the provisioning of ecosystem goods and services[4–7].

According to Diffenbaugh and Giorgi[8], Central America counts among the global climate change hotspots in view of projected increasing mean temperatures, more frequent extreme temperature events and higher interannual precipitation variability. Concurrently, the region is listed as a global hotspot of biodiversity and hosts more than 2900 endemic plant species[9]. Central American forest ecosystems in particular serve a critical role as habitat for rare species and also represent an increasingly popular destination for ecotourism. Overall, these ecosystems are expected to provide high levels of goods and services[10–12]. In contrast, land scarcity limits the available space for natural forests, which are in conflict with a high demand for tropical plantation forests as timber source or green investment for carbon sequestration[13]. Additionally, Central America's distinct geography imposes physical limitations on species distributions: the dry Mexican highlands constitute a natural barrier in the north, the isthmus of Panama borders the south and scattered volcanoes sit atop the higher mountain ranges[14,15] (also see Fig. S26). In conjunction with the high spatial heterogeneity of the landscape, distinct forest types have evolved, which feature tree species that are closely adapted to the local environmental conditions and follow specific resource use strategies. Due to this high specialization and the scarcity of alternative habitats, however, tropical biodiversity is expected to be particularly sensitive to climate change[16]. Range shifts of suitable habitat could therefore lead to high risks of habitat connectivity loss or—following upslope shifts towards elevation peaks—mountaintop extinction[17].

To investigate potential range shifts under different climatic states, species distribution models (SDMs) represent a commonly applied tool[18,19]. State-of-the-art SDM techniques commonly build ensembles of multiple individual models to cover a broader range of algorithms and improve overall robustness of model predictions[20]. In the Central American region SDMs have been occasionally used before[21,22], yet their results are limited to single species. For better conservation planning, however, an extension of such focus studies beyond the species level is desirable. A simple way forward lies in the aggregation of many single-species models to stacked-SDMs (SSDMs) to summarize predictions across species and gain insights about more general trends in species communities[23,24]. Nevertheless, for investigating species communities or ecosystems pure "mass stacking" SSDM approaches are impractical, since they require large computational efforts and multiply uncertainties of individual SDMs. Therefore, in contrast to such species-specific modelling, trait-based approaches have been increasingly gaining attention in ecological research. By focusing on the relationship between physiological, morphological and life-history characteristics of organisms and their environment, these approaches allow for the identification of more fundamental patterns beyond the species level[25]. Consequently, interlinking SDMs with trait-based approaches may be particularly valuable for analyses in species-rich regions such as Central America.

Here we investigate how changing climatic conditions will influence the environmental suitability of main tree plant functional types (PFTs) in Central America. Therefore, we grouped widespread regional tree species into seven PFTs: wet acquisitive, wet conservative, dry acquisitive, dry conservative, coniferous, montane and generalist. For each type we trained stacked species distribution models using multi-model ensembles and a collection of more than 20,000 species occurrence records. We then projected the environmental suitability of each PFT at a high spatial resolution (30-arc seconds, ~1 km$^2$) under nine climate change scenarios for 2061–2080, combining three representative concentration pathways (RCP; 2.6, 4.5 and 8.5) with three global circulation models (GCM; CCSM4, HadGEM-AO and MPI-ESM-LR). We particularly analysed our results for shifts of core areas of suitability along latitude and altitude. Further, we identified potential transitional areas between PFTs and mapped fragmentation to visualize threats of habitat connectivity loss or isolation, which could promote species extinctions. Finally, we discuss implications for conservation ecology and the provisioning of forest ecosystem services.

## Results

**Dominant PFTs.** To highlight core areas of suitability and detect potential transition areas between forest types under the influence of climate change, we determined the PFT with the highest predicted occurrence probability for each 30-arc-second grid cell in our study region (total area: 1,846,817 km$^2$). For easier description, we termed this "dominance". This only refers to the suitability of environmental conditions and should not imply any competitive advantage. Figure 1 illustrates these results spatially (panel a), shows trends for each PFT across RCPs (panel b) and also highlights flows between types (panel **c**) for the CCSM4 scenarios (for other GCMs see Figs. S1, S2, Supplementary). Under current climate conditions, the dry forest PFTs showed the highest occurrence probabilities in the north (especially Yucatan peninsula) and along the Pacific coast of Central America, transitioning from conservative to acquisitive types from north to south. In contrast, wet forest PFTs dominated the lowlands facing the Caribbean, with a trend from conservative in the coastal areas to acquisitive towards the inland and premontane areas. Montane species prevailed in the high mountain ranges of the American Cordillera between the wet-dry forest frontier and coniferous species in the lower montane subtropical areas (>12°N).

Throughout the mild (RCP2.6) climate change scenarios, these patterns remained largely constant for all realized GCMs. More pronounced dominance shifts between PFTs appeared under moderate climate change scenarios (RCP4.5). Areas along the Caribbean coast changed from predominantly wet conservative species under present-day climate to be more suitable for the wet acquisitive and generalist PFT. Dry forest dominated areas did not shift remarkably. Under RCP 8.5, the previously described trends were reinforced. Areas most suitable for generalists extended into large portions of Nicaragua, Costa Rica and Panama at the expense of the wet PFTs. The present-day dominance of montane and coniferous species in medium to high altitudes declined with increasing climate change intensity in an almost exponential manner. Notably, some regions in the centre of the study area did not show high occurrence probabilities for any PFT. In Nicaragua and Belize these largely persisted throughout all scenarios and could point to areas possibly more suitable for PFTs not considered in this study, whereas in Honduras and Guatemala these areas increased with increasing RCP strength and may indicate new combinations of environmental conditions under climate change.

**Latitudinal and altitudinal shifts.** An analysis of directional shifts of predicted presences along latitudinal and altitudinal gradients revealed additional patterns across PFTs. A type was considered present when the sum of the binarized stacked model predictions (bS-SDM) met the threshold ≥ 2. Figure 2 shows these

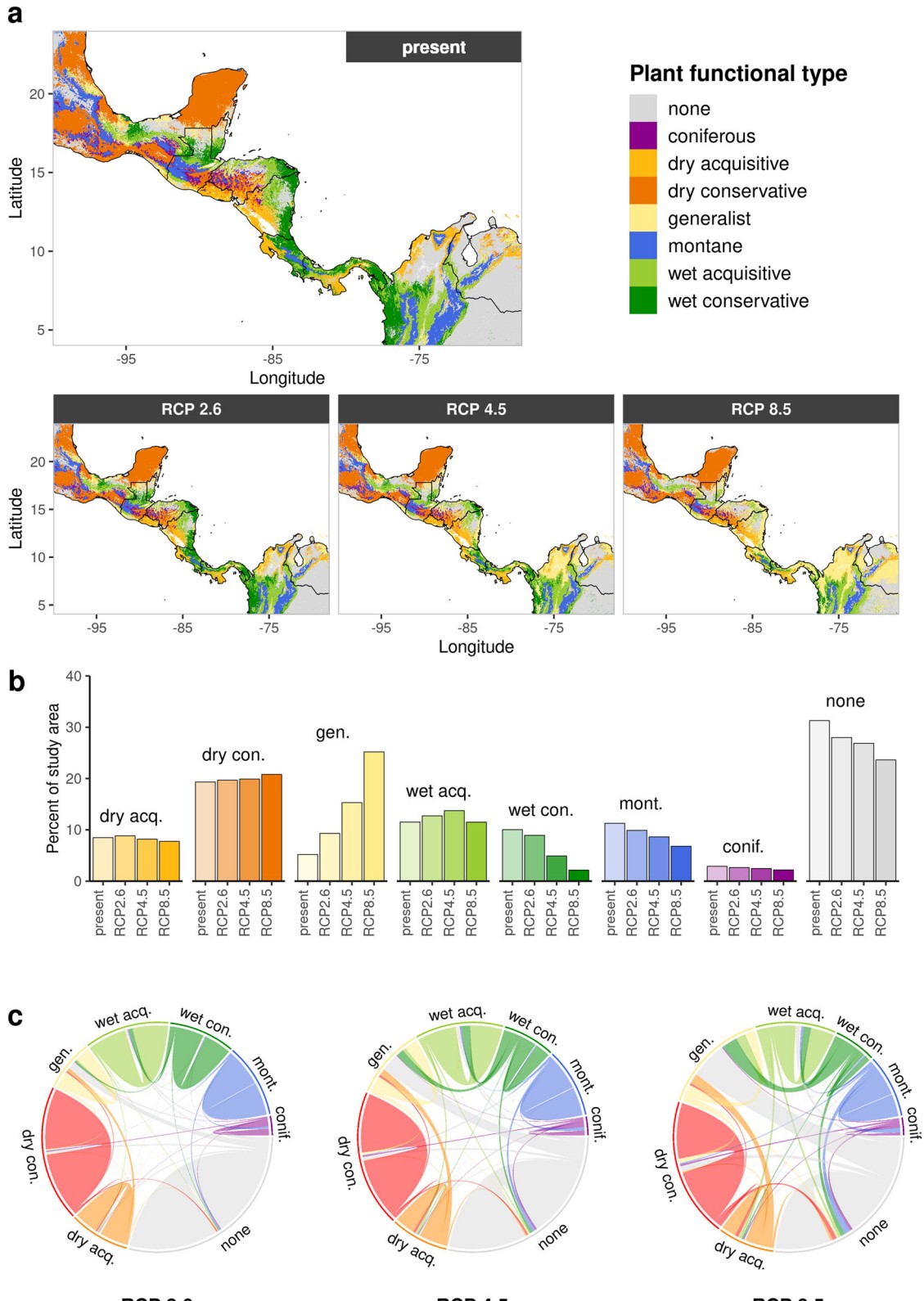

**Fig. 1 Dominant PFTs.** Model projections for the CCSM4 scenarios. **a** Maps showing the dominant PFT (type with the highest occurrence probability) for each grid cell. "None" refers to the case, where fewer than two species were predicted as present for every type (bS-SDM < 2). **b** Relative area covered by each dominant type (area statistics of **a**). **c** Chord diagram illustrating flows between dominant types for RCP 8.5 compared to present. All (**a**), (**b**) and (**c**) share the colour code of the PFT legend.

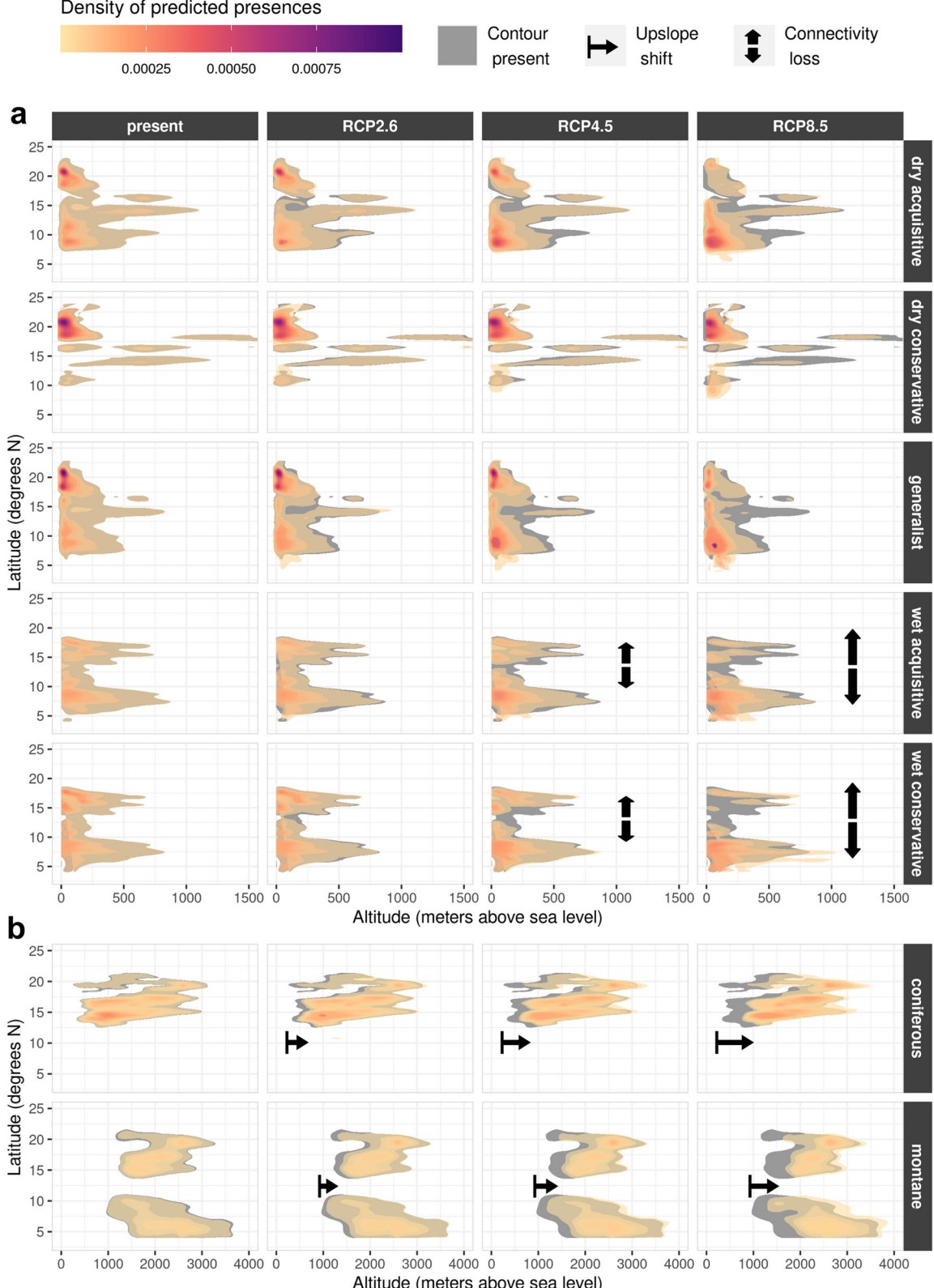

**Fig. 2 Latitude-altitude suitability shifts.** Density plot of projected PFT presences (bS-SDM ≥ 2) across latitudinal and altitudinal gradients (**a**: lowland types, **b**: coniferous/montane types). Columns 2-4 show CCSM4 scenario projections with grey shading of the contour of column 1 (present). Accordingly, fully visible grey contours highlight zones, which may be lost in the respective scenarios. Diverging arrows mark scenarios where connectivity losses appeared (arrow length symbolizes strength), while right-facing arrows represent the strength of upslope shifts of the lower boundary of the density clouds (trend towards mountaintop extinction).

outcomes in the form of density clouds of the predicted presences for the CCSM4 scenarios (for other GCMs see Figs. S3, S4, Supplementary). For better readability, the results are displayed separately for the lowland types (panel a: dry, wet, generalist) and montane/coniferous types (panel b) and highlights potential habitat connectivity losses and upslope shifts (as an indicator for the threat of mountaintop extinction). Most notably, the lower boundary of montane and coniferous species suitability centres shifted upwards by approximately 100 m for RCP2.6, 200 m for RCP4.5 and 500 m for RCP8.5. On the other hand, the highest densities of dry acquisitive and generalist species presences gradually shifted from north to south, whereas dry conservative species showed slightly increased densities between 8–12°N. The wet forest PFTs also trended towards the south, but diverged into two or more separate density centres under RCPs 4.5 and 8.5. Additionally, small upwards shifts could be noted for the wet conservative type.

**PFT fragmentation**. To investigate potential connectivity issues in geographic space, we mapped the fragmentation class for each grid cell and PFT as calculated from a 3 × 3 moving window analysis. The results for the CCSM4 scenarios are shown in Fig. 3 (for other GCMs see Fig. S5 & S6, Supplementary). In addition to predicted losses of up to 78% for the wet conservative PFT (compare Fig. 1), its remaining areas showed increasing proportions of fragmented landscapes (patches and transitional areas). These moderately increased for the mild scenarios but almost doubled for RCP8.5 in comparison to present (for details see Table S1). While the geographic distance between suitable areas along the Caribbean coast was already large under present-day projections, it increased exponentially with RCP strength, leaving the remaining forest patches largely isolated. The total area dominated by the wet acquisitive PFT moderately increased for all but one scenario. Despite overall small increases of the interior area fractions, several connectivity bottlenecks appeared along the Caribbean coast of Honduras, Nicaragua and at the isthmus of Panama. These gaps were mainly filled by the generalist PFT, which multiplied its total area by 2 (RCP2.6) to 5 (RCP8.5) and also doubled its interior portion. Both dry forest PFTs on the other hand only showed slight total area increases. Fragmentation did not change for the dry conservative PFT, but for the acquisitive type a moderate trend towards less interior portions and higher fractions of patches, transitional and perforated areas was found. Coniferous areas decreased slightly, but did not influence the partitioning of fragmentation classes. Finally, albeit areas covered by the montane PFT shrunk by up to 44%, this only led to very weak decreases of the interior and increases of the perforated fraction.

## Discussion

Our model projections for 2061-2080 support existing studies on Central American forests with regards to a general trend towards dry vegetation types[4,6,26]. Beyond these generalities, however, this study also found regional disparities and PFT-specific responses, which may be attributed to small-scale heterogeneity of climate, topography and soils[27]. Most PFTs showed gradual shifts of suitable area towards the south, which could lead to a concentration of competing species and - due to the low geographic connectivity between Central and South America - a rigid distribution limit. At the same time diverging latitudinal and altitudinal trends of suitable area for wet forest species may impair habitat connectivity and reinforce fragmentation effects caused by human land use. Particularly, connectivity bottlenecks appeared in the Mesoamerican biological corridor along the Caribbean

coast, which is an essential North-South migration and dispersal route[28,29]. Increasing proportions of highly fragmented forest patches may further reduce the available forest interior area which many species depend on[30,31]. For coniferous and montane species, suitable areas showed upwards trends, but only at the lower distribution boundary. Since these types in most cases have already reached the highest areas they may be facing mountaintop extinction in the long term[32].

These developments could have dramatic implications for Central America's biodiversity. Compared to the other types, wet and montane forests host the largest number of amphibians, birds, mammals and reptiles by far, many of them already under threat of extinction or suffering from habitat fragmentation[10]. To protect wildlife and maintain the integrity of these ecosystems, it is thus becoming increasingly important to delineate biological corridors and include them into land development plans (also compare Fung et al.[33]). Equally, for species facing mountaintop extinction due to the lack of alternative habitats, there is an urgent need to create or extend protected areas to act as refugia. Besides this conservation ecology perspective, transitions from wet to dry forests would also affect ecotourism, which is mainly centred around rain forests and constitutes an important source of income, particularly in Costa Rica and Panama[34]. Additionally, the lower assimilation and growth rates and consequently reduced carbon sequestration of dry forests could entail economic impacts[35,36]. Particularly pine species, which are widely used for timber production, could be in strong competition with species better adapted to drought. Recurring pest outbreaks and El Niño induced dry spells may further exacerbate this development and have already caused substantial economic damages[37]. Lastly, beyond the here modelled vegetation types, other plant groups such as lianas, palms or grasses could also alter forest structure and growth due to their role in gap dynamics after forest disturbances[38–40]. For instance, recent trends of increasing liana biomass across tropical forests could be reinforced under stronger global warming, resulting in reduced tree growth and overall reduced forest carbon uptake[41,42]. Exploring the competitive relationship between these plant groups from a vegetation growth perspective under the influence of climate change and different disturbance regimes thus represents an important avenue for future research.

In contrast to the aforementioned bleak prospects, recently increasing net reforestation in Central and South America at least points to increasing efforts to restore forest ecosystems[43]. In combination with the conservation of climatically stable areas and key areas for landscape connectivity as highlighted in this study, climate change impacts on remaining natural habitats may be bolstered and biological corridors maintained.

Overall, our study revealed both general trends and hotspots of forest type transitions in Central America with regards to their climatic suitability under different climate scenarios. In particular, decreasing climatically suitable area for wet forest species, impaired habitat connectivity and mountaintop extinctions emerged as possible threats and highlight the need for urgent policy interventions. To extend these findings, further research on growth responses to climate change throughout different biomes and vegetation types is needed to better inform management decisions (similar to Stan et al.[44]). For biodiverse regions such as Central America, trait-based approaches as used in this study may represent a valuable perspective to combine ecological understanding with practical application (e.g. trait-based species selection). Finally, the application of similar modelling approaches to other biodiversity hotspots could assist in identifying the evolving threats of connectivity loss and mountaintop extinctions globally and safeguarding endangered habitats.

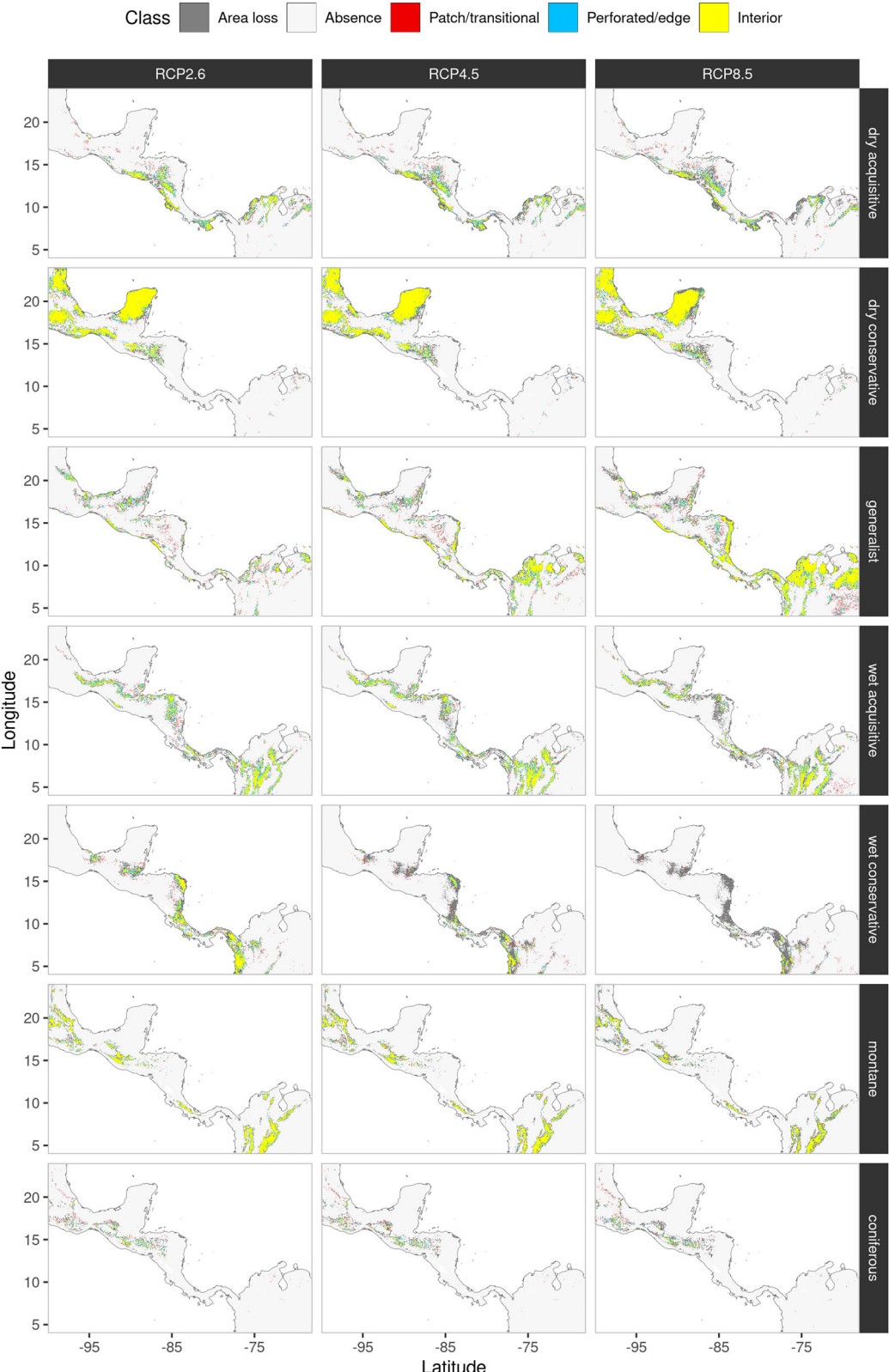

**Fig. 3 PFT fragmentation.** Maps showing fragmentation classes for each PFT in the areas where they were predicted as dominant (CCSM4 scenarios). Dark grey marks areas, which were covered by the PFT in present-day projections, but disappeared in the scenarios ("Area loss"). For definitions of the other fragmentation classes see Methods section.

## Methods

**Study area**. The study region covered the Central American subcontinent between the longitudes 68°–100°W and the latitudes 4°–24°N (excluding the Caribbean islands). This extent included parts of the neighbouring biogeographic regions of Mexico and Northern South America, which extended the calibration range of our models to be better suited for projections to different climates.

**Species selection and grouping**. For the definition of PFTs we considered both abiotic tolerances and structural/physiological plant attributes. As a main biogeographic division we followed Olson et al.[45] and Corrales et al.[10] and distinguished between wet and dry forest in the lowlands (<1000 m) and coniferous and (broadleaved) montane forests in higher altitudes (for a topographic overview of the study region see Fig. S26). In addition, we considered different resource acquisition strategies, reflected in specific functional traits, to further group wet and dry forest species into acquisitive and conservative types[46]. While acquisitive traits comprise attributes necessary for high assimilation rates and fast growth, conservative traits aim at resource preservation and increased stress tolerance (for details see Supplementary methods). Since some Central American tree species show trade-offs between acquisitive and conservative traits and occur widespread across dry and wet biomes, we grouped them into a separate "generalist" type. The final set of PFTs included: dry acquisitive/conservative, wet acquisitive/conservative, coniferous, montane and generalist species. For the representation of each PFT we selected four tree species based on extensive literature review[22,47–50], trait data analysis[48,49,51–62], broad occurrence within respective ecoregions (i.e. not region-specific, see Fig. S27) and a minimum availability of at least 200 species records (compare Soultan & Safi[63]). A summary of the selected species for each PFT, mean trait values and sample sizes is shown in Table 1.

**Data**. Species presence records were obtained from a large number of datasets within the collections of the Global Biodiversity Information Facility[64], the Botanical Information and Ecology Network (BIEN[65–69]), and de Sousa et al.[70]. GBIF and BIEN data were retrieved via the R packages rgbif[71] and BIEN[72]. Where synonymous taxa occurred, these were assigned to the respective accepted names of the GBIF backbone taxonomy. Due to common confusion and close co-occurrences of Weinmannia species[73], we aggregated all Central American occurrences into a species group (Weinmannia spp.). For all species, only records falling onto land within the study area were kept. To reduce sampling biases, we additionally thinned the presence data down to one record per cell on the grid of the environmental predictor data.

We preselected a set of predictor variables based on known species characteristics and environmental limits[48,49,52,61,62]. The data were obtained at 30-arc-second resolution from different sources: bioclimatic variables from the CHELSA dataset[74] (version 1.2), topographic data from GMTED2010[75] and edaphic variables from SoilGrids1km[76]. Prior to modelling, we tested all obtained variables for multi-collinearity following a supervised step-wise procedure, where the variable with the highest variance inflation factor was excluded in each step until all variables ranked <10. The final set of predictors comprised: maximum temperature of the warmest month, minimum temperature of the coldest month, precipitation seasonality, annual precipitation, soil sand fraction (30 cm depth), soil clay fraction (30 cm depth), soil pH in water (30 cm depth), depth to bedrock and hillslope. For the climatic variables, we obtained present-day data (1979-2013) and scenario projections from three global circulation models (GCMs; CCSM4, HadGEM2-AO and MPI-ESM-LR) and for three representative concentration pathways (RCPs; 2.6, 4.5 and 8.5). A comparison between projected and present-day annual precipitation is shown in Fig. 4 (for other variables see Fig. S7–S10).

**Modelling approach**. Models were built using the R package SSDM[77]. To account for algorithm biases[78], we tested a broad spectrum of commonly used SDM algorithms to be used as model ensemble[79–81]. This included artificial neural networks (ANN), classification tree analysis (CTA), generalized additive models (GAM), generalized linear models (GLM), multivariate adaptive regression splines (MARS), Maximum Entropy (MAXENT), support vector machines (SVM) and random forest (RF). Modelling settings and strategies for pseudo-absence selection were customized for each algorithm based on suggestions by Barbet-Massin et al.[82] (for details see Supplementary methods). Per algorithm and species 100 model replicates were trained and evaluated through cross-validation by splitting the occurrence data into 70% training and 30% holdout test data. For a full description of our settings also see the ODMAP protocol provided in the Supplementary methods.

The raw output of the resulting SDMs is the probability for a given species to occur under the given environmental conditions. Following the model training, the SDMs were first evaluated for their discrimination capacity using the Area under the curve (AUC). Since we aimed for projecting suitability over space and time, we also evaluated model calibration by using the Continuous Boyce Index (CBI[83]) and - except for MAXENT (as a presence-only method) - a novel calibration statistic that summarizes the calibration plot (sdm package[84]). We required all models to meet a minimum threshold of 0.7 for these evaluation metrics. For most species ANN, CTA, GLM and SVM models fell below the threshold, so that we excluded

these methods from our analysis. The remaining GAM, MARS, MAXENT and RF models were projected to new environments by updating the climatic variables with scenario data. Since future climatic conditions may not find an analogue in the climatic conditions of the training data, model extrapolation can become a problem. Within our focus region only small areas showed non-analogue values and no interaction of novel conditions occured between variables (see Fig. S11). We thus decided to clamp values exceeding the training range to the minimum/maximum values present in the study region to avoid extrapolation altogether. Predictions in the clamped areas still need to be treated with caution, since they may underestimate climate change effects. To investigate main sources of uncertainty between model projections, we ran an analysis of variance (ANOVA) for each grid cell and species with respect to the factors algorithm, training dataset (model replicates), RCP and GCM (Fig. S24). Among these, algorithm choice by far explained most variance, followed by RCPs and GCMs, while model replicates had the lowest impact. This is in agreement with Diniz-Filho et al.[85]. For the subsequent steps, we thus selected only 10 replicates per algorithm and species to improve computational efficiency.

While model ensembling provides the opportunity to decrease the impacts of algorithm biases and emphasize commonalities, it requires well-calibrated model projections to be interpreted on a common scale as absolute occurrence probabilities[86]. To this end, we tested whether recalibration of our model projections could improve calibration (as suggested by Phillips and Elith[87]). All recalibration methods included in the calibratR package[88] were applied for this purpose. However, calibration errors did not improve notably and CBI even decreased for most methods (Table S3). We thus ensembled the original model projections as unweighted arithmetic means for each species across algorithms and replicates. For each PFT we then summed probabilities of the corresponding four species. The resulting "richness maps" were used as indicators to compare relative suitability across climate scenarios and PFTs. For additional explorative analyses, we also computed sums of the binarized model predictions (bS-SDM) using the maximum sum of sensitivity+specificity as binarization threshold (compare[89]). Only grid cells where at least three of the algorithms agreed were considered presences in each species ensemble. Further, to reduce the sensitivity towards single species outliers, we only counted cells with bS-SDM ≥ 2 as PFT presences (for detailed results see Supplementary, Fig. S12–S25). With these results, we finally investigated potential geographic range shift trends as projected by the models by plotting the density of presences over latitude and altitude.

**Model evaluation**. The individual SDMs showed overall good performance for all species (see Fig. 5). On average, random forest models achieved the highest values for discrimination and calibration criteria and had the lowest variance across models. Unsurprisingly, models for narrow-niched species such as the montane type achieved the highest AUC values (compare[90]). The broad-niched generalist type models showed a trade-off between comparably lowest AUC values but highest calibration performance. Remarkably, rankings of discrimination and calibration performance by PFT were very similar between the algorithms. We assume this could be a possible effect of the niche complexity or the level of appropriateness of the predictor variables for the different PFTs.

**Model limitations**. The use of SDMs for climate change studies is often contested due to the cascade of uncertainties that comes in their wake, e.g. disequilibrium of occurrences and predictors, sampling bias, spatial autocorrelation or algorithm bias (for a detailed discussion see Supplementary methods). While we generally agree with these concerns, we would like to stress, that the largest weakness of SDMs does not originate from wrong conceptualization, but from wrong application. Here, we limit our interpretation to exploring general patterns and put special emphasis on assessing and communicating uncertainties. In this form, we still see substantial value of SDMs to contribute to our understanding of the environment, particularly when time and resources are limited.

For the interpretation of our results, it should be noted that environmental suitability maps as projected by SDMs can only point to the favourability of environmental conditions and are based upon an assumed equilibrium state of species occurrences with training environmental conditions. They may especially not give a time line or "expiration date" for species range shifts or extinction. Accordingly, even if no suitable habitat would be projected for a given species under future climatic conditions, an "actual extinction" could be delayed or sped up by a large number of additional factors[91]. On a physiological level, survival and growth of plants are strongly influenced by disturbances (droughts, fire, pests, temperature extremes), light and nutrient availability, $CO_2$ concentration, dormancy or dispersal strategies, which may inhibit the actual occurrence of a species at a given site[36,92,93]. In addition to this, migration limits may apply, either due to biogeographic barriers such as islands, mountains (e.g. American Cordillera) and topographic bottlenecks (e.g. Isthmus of Panama) or anthropogenic structures and land use. Particularly in view of Central America's increasing urbanization, cropland expansion and biogeographic limits the actually available area for range shifts is drastically reduced.

**Fragmentation analysis**. For the analysis of PFT fragmentation patterns and potential connectivity bottlenecks across the landscape, we used a classification approach by Riitters et al[94]. Due to its categorical output, the results are easily

**Table 1 PFT summary.**

| PFT | Drought tolerance | Altitudinal zone | Shade tolerance | Wood density | Leaf nitrogen | Specific leaf area | Seed dry mass | Leaf phenology | Representative species | Sample size |
|---|---|---|---|---|---|---|---|---|---|---|
| Dry acquisitive | ◄ | ▼ | ▼ | 0.56 ± 0.07 | 2.35 ± 0.37 | 13.66 ± 0.73 | 525.20 ± 314.35 | D | *Brosimum alicastrum*<br>*Calycophyllum candidissimum*<br>*Enterolobium cyclocarpum*<br>*Pachira quinata* | 1316<br>550<br>905<br>334 |
| Dry conservative | ◄ | ▼ | ▼ | 0.65 ± 0.06 | 2.69 ± 0.56 | 11.98 ± 1.10 | 37.00 ± 20.31 | SE/E | *Alvaradoa amorphoides*<br>*Byrsonima crassifolia*<br>*Leucaena leucocephala*<br>*Vachellia farnesiana* | 641<br>2340<br>1074<br>1125 |
| Wet acquisitive | ▶ | ▼ | ▼ | 0.32 ± 0.05 | 2.57 ± 0.31 | 16.54 ± 3.60 | 281.825 ± 266.76 | SE/E | *Cecropia obtusifolia*<br>*Ochroma pyramidale*<br>*Schizolobium parahyba* | 686<br>660<br>348 |
| Wet conservative | ▶ | ▼ | ◄ | 0.63 ± 0.08 | 1.96 ± 0.26 | 14.22 ± 0.67 | 8789.65 ± 6440.16 | SE | *Vochysia ferruginea*<br>*Calophyllum brasiliense*<br>*Carapa guianensis*<br>*Dialium guianense*<br>*Symphonia globulifera* | 432<br>760<br>314<br>489<br>599 |
| Generalist | ▲ | ▼ | ▲ | 0.42 ± 0.06 | 2.28 ± 0.08 | 15.93 ± 1.21 | 745.93 ± 539.51 | D/SE | *Ceiba pentandra*<br>*Guazuma ulmifolia*<br>*Simarouba amara*<br>*Spondias mombin* | 810<br>3225<br>498<br>1520 |
| Montane | ▶ | ◄ | ▲ | 0.49 ± 0.03 | 2.08 ± 0.29 | 8.38 ± 1.31 | n/a | SE/E | *Alnus acuminata*<br>*Cornus disciflora*<br>*Drimys granadensis*<br>*Weinmannia spp.* | 1115<br>657<br>822<br>1468 |
| Coniferous | ◄ | ▲ | ▼ | 0.50 ± 0.02 | n/a | n/a | 28.90 ± 13.19 | E | *Pinus ayacahuite*<br>*Pinus caribaea*<br>*Pinus oocarpa*<br>*Pinus tecunumanii* | 270<br>206<br>618<br>257 |

Summary of the PFT composition (species names in italics), functional traits and sample sizes (presence records). Trait means are shown in bold with respective standard errors indicated by ±. Qualitative traits are ranked as high/intermediate/low (▲/▶/▼). Leaf phenology can be deciduous (D), semi-evergreen (SE) or evergreen (E).

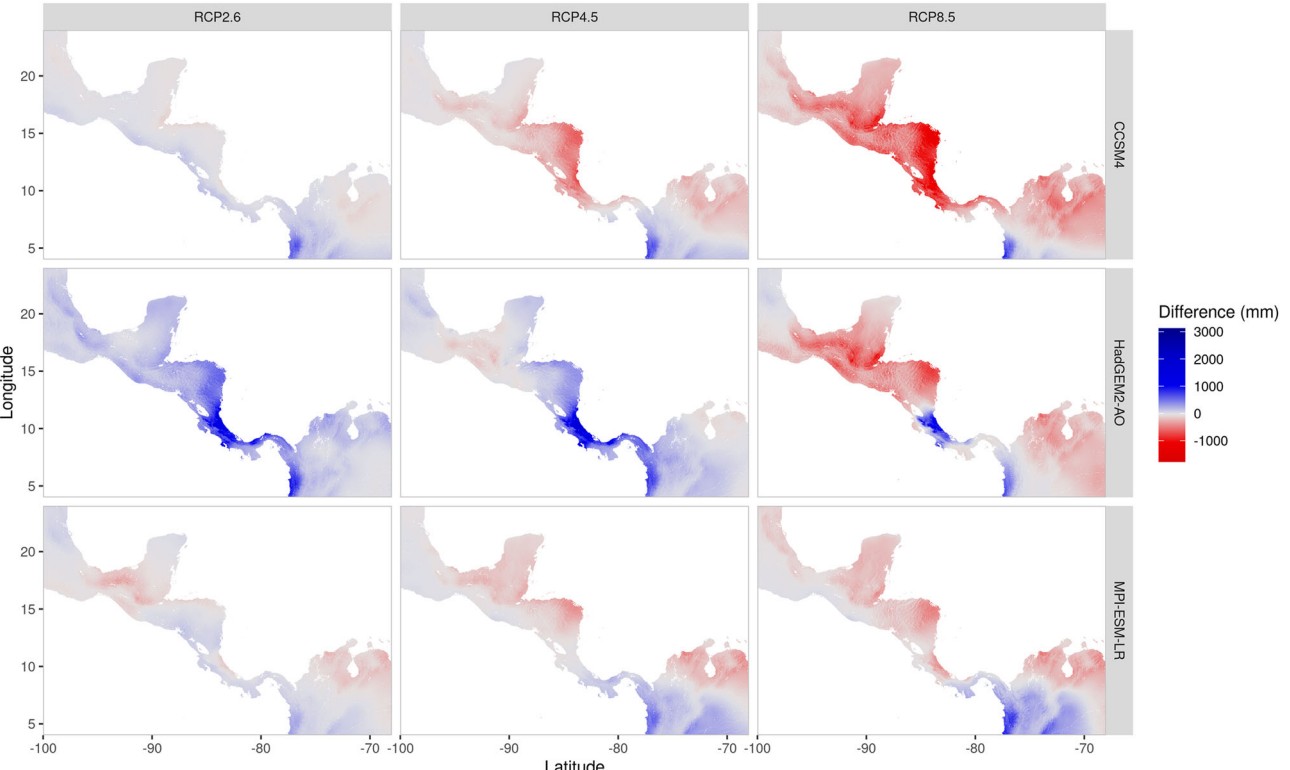

**Fig. 4 Precipitation change.** Difference of annual precipitation sum between present-day and climate projections (2061–2080).

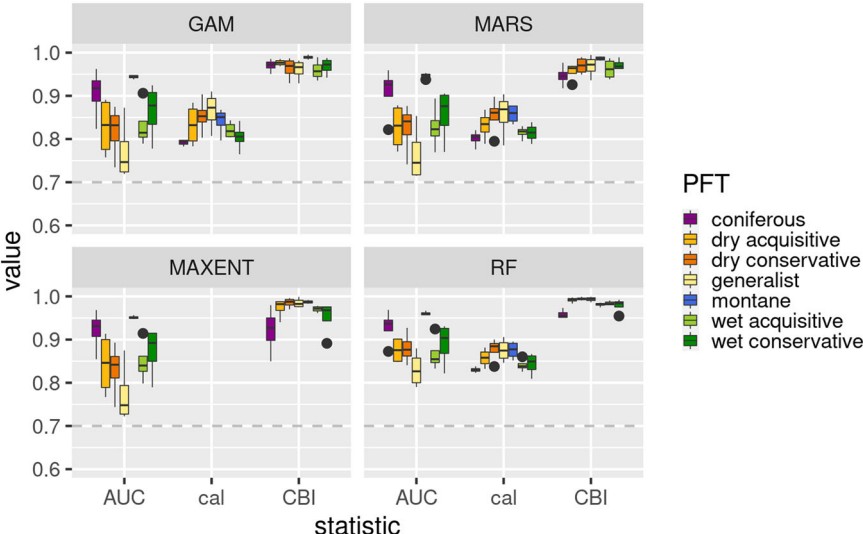

**Fig. 5 Model evaluation.** Summary of the evaluation metrics Area under the curve (AUC), calibration statistic (cal) and Continuous Boyce Index (CBI) for each algorithm (GAM = Generalized Additive Model, MARS = Multivariate Adaptive Regression Splines, MAXENT = Maximum Entropy, RF = Random Forest) and by species group. The dashed line represents the applied model selection threshold (0.7).

interpretable within a conservation context, as has for example been successfully shown by Morelli et al.[95]. In the first step we created binary maps for the presence (value = 1) of each PFT from the "dominant PFT" maps (Fig. 1). Subsequently, within a 3 × 3 moving window we calculated 1) the proportion of cells covered by the respective PFT ("Pf") and 2) the conditional probability that a cell with a value of 1 has a neighbour cell with a value of 1 (connectivity, "Pff"). All calculations were realized with the fasterRaster R package[95]. Both Pf and Pff were compared to derive fragmentation classes: (1) interior, if Pf = 1.0; (2) patch, if Pf < 0.4; (3) transitional, if 0.4 < Pf < 0.6; (4) perforated, if Pf > 0.6 and Pf - Pff > 0; (5) edge, if Pf > 0.6 and Pf – Pff < 0 or if Pf > 0.6 and Pf = Pff.

**Reporting summary**. Further information on research design is available in the Nature Research Reporting Summary linked to this article.

## Data availability

CHELSA v1.2 bioclimatic gridded data can be downloaded from https://chelsa-climate.org/downloads/, GMTED2010 elevation data from https://edcintl.cr.usgs.gov/downloads/sciweb1/shared/topo/downloads/GMTED/Global_tiles_GMTED/300darcsec/mea/ and soil data from https://soilgrids.org/. Tree physiological data can be largely obtained from literature[48,49,52,55–62]. Additionally, access to restricted trait datasets[51,53,54] may be

requested from Diego Delgado (CATIE) and Robin Chazdon (University of Connecticut). The collection of species occurrence data was assembled from the databases of the Global Biodiversity Information Facility[64], the Botanical Information and Ecology Network[65–69], and de Sousa et al.[70]. The compilation is available from https://doi.org/10.5281/zenodo.4835834 [96]. All data generated in this study are available from the corresponding author on reasonable request. Source data underlying graphs in the main figures is available from the Supplementary Data (https://doi.org/10.5281/zenodo.4836270)[97].

## Code availability
The R packages 'SSDM' (version 0.2.8), 'ENMTools' (version 0.2) and fasterRaster (version 0.6.0) are available on GitHub (https://github.com/sylvainschmitt/SSDM, https://github.com/danlwarren/ENMTools, https://github.com/adamlilith/fasterRaster), 'calibratR' (version 0.1.2) is available on CRAN (https://cran.r-project.org/web/packages/CalibratR/index.html). The 'circlize' R package (version 0.4.1)[98] used to prepare circular subplots can be obtained from GitHub (https://github.com/jokergoo/circlize). The R code underlying our simulations is available from https://doi.org/10.5281/zenodo.4835834 [96].

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

## Acknowledgements

This work has been financially supported by the German Research Foundation (DFG), 416575874, HA5971/2-1. We acknowledge the following herbaria that contributed data to this work: CVRD, FURB, IAC, INPA, IPA, MBML, UESC, UFRN, UFS, US, USP, BRIT, MO, NY, TEX, U. We particularly thank the Unit of Forests and Biodiversity in productive landscapes at CATIE (Costa Rica) for granting us access to the local species trait databases and sharing their expertise in forest ecology. Finally, we acknowledge Vanessa Boukili and Robin Chazdon (projects funded by the National Science Foundation NSF DEB 1147429, 1050857 and 1110722) for contributing valuable trait data.

## Author contributions

L.B., R.Y. and M.H. conceived the idea. L.B. and D.W. elaborated the methodological framework. L.B. performed the analyses and prepared the figures. L.B. prepared the first draft of the manuscript. D.W., R.Y. and M.H. substantially contributed to further revisions.

## Funding

## Competing interests

The authors declare no competing interests.
