## [Transparent Peer Review File · Communications Biology]

Reviewers' comments:

Reviewer #1 (Remarks to the Author):

The manuscript entitled "Climate change may induce connectivity loss and mountaintop extinction in Central American forests" predicts the response of species trait-based types to multiple environmental stressors using three different RCP scenarios in Central America (Guatemala to Panama). In summary, the authors found that vegetation adapted to wetter climates would most likely be substituted by dry adapted and/or generalist vegetation. Model predictions also showed that with less mobility latitudinally, wet adapted species would lose connectivity. If they moved up in elevation, they were at risk of mountaintop extinction as climate becomes more severe (i.e., warmer and drier). This work is interesting and relevant since it directly addresses vegetation shifts due to climate change, in an area that indeed harbors a large number of endemic species in addition to containing very heterogeneous terrains and ecosystems. I see this manuscript being of interest to a wide range of researchers (i.e., plant ecologists, climate modelers, hydrologists, etc).

The study is straightforward, the manuscript is well written, and I appreciate the authors acknowledging the downfalls of the models used. However, I am struggling to find the novelty of this manuscript. As written by the authors in the Discussion, their findings mirror the findings of other studies. They defend that their study focuses on a smaller-scale view of vegetation response to climate change. However, the number of species used (Table 1) for each "species type" was quite low for the size of the area that they were simulating. I acknowledge all the work put in to classifying each species based on prior literature, online databases, and herbaria; but I do worry that some of the species type-specific shifts found might be incorrect. I believe the dry vs. wet adapted plants, and pine results are correct, but it becomes a little more complicated when considering the other types.

Similarly, as highlighted by the authors, some vegetation were not classified into any of the categories, and thus not considered in the result analysis. However, considering the forest dynamics of tropical environments, I found that excluding lianas and palm trees (for example) from the predictions could have biased their results. I highlight these two plant functional groups specifically due to their importance in the tropical stand composition (i.e., palm trees as hyperdominants in the Amazon, but also frequently found in Central America) and their colonization/invasive nature when extreme events occur (i.e., lianas colonize forest gaps and sometimes dominate to the point of exclusion of local trees). Thus, I wished these groups (and possibly others) were also considered in these simulated scenarios.

Finally, I did not agree with the use of "species types" throughout the text. In my opinion, there isn't a "type" of species if that is not based on a criteria. In this case it was "traits". So, I strongly recommend switching throughout "species types" for "species trait-based types" or "plant functional groups". Please refer to the revised manuscript PDF attached for more details.

Reviewer #2 (Remarks to the Author):

This study examines outcomes for trait-based climate change distribution modeling for tree species in Central America. The main findings are that there will be transitions from wet towards generalist or dry forest tree species over a large portion of the study region. Additionally, the authors found that predicted suitable area for wet forest species will latitudinally diverge and lose connectivity, and that upslope shifts of suitable areas for montane species may increase risk for mountaintop extinction. Though the use of modeling for forest species distributions in differing climate envelopes, even within this study region, is not novel (see. Lyra et al. 2017, Lyra et al. 2016, Imbach et al. 2012), the execution of the modeling methodology is sound, and the findings are relevant and important for conservation planning. The species-type/trait-based guilds used in these analyses have been performed for climate distribution models (Diamond et al. 2012) but are novel for the region and an

ecologically relevant and important way to model climate scenarios across landscapes for large-scale conservation planning.

Areas in need of improvement include adding SDM and ensemble SDM climate modeling, as well as trait-based distribution modeling background to the introduction. Currently there is no background or reference to distribution modeling for species across climate scenarios in the introduction, though there is a large literature on the topic. The authors have included one sentence addressing previous similar studies conducted in the region in the discussion, but a sentence or two discussing SDMs used for climate models of species distribution shifts in the introduction would be beneficial for those unfamiliar with the topic.

Additionally, I was unable to find sample sizes for models provided in the main text or supplementary material. Adequate sample sizes are important for these types of models and n should be included in the manuscript. The numbers of background and pseudo-absence points were included in the supplementary materials (SM), but authors should consider including them within the main article as they are important for assessing the soundness of the models. However, many of my initial questions and issues with the manuscript were addressed in the SM, including uncertainty maps for the models and the detailed SM is appreciated.

The manuscript could also benefit from more clarity on the posited connectivity loss as connectivity loss is emphasized as a key outcome in this study. Specifically, how are the strengths of divergence measured (what metrics used) then visualized for Figure 2? The authors posit that due to shifts and divergences in predicted population densities, there would be loss of connectivity. Though I agree connectivity loss is likely, there is no analysis, example, nor background (sources) presented to support this claim. Similarly, the author's mountaintop extinction outcome is in need of support, either through referenced sources or metrics from their study. Sources and background could be included in the discussion of implications of their study.

Overall, this study is thought out and the methodology is well designed. The manuscript itself could be bolstered with more detailed background and references. A check for awkward phrasing throughout the manuscript would be helpful as well. Specific comments below:

Abstract

- line 23: delete "the" before "identified transition"

Introduction

- line 30: delete "Over the past decades"
- lines 38-40: awkward phrasing – restructure
- line 40: "Concurrently" to replace "At the same time"
- lines 42-44: awkward – restructure
- lines: 67-70: awkward
- lines 72-79: This paragraph should be in Results section, not in the Introduction section

Results

- line 83: was "suitability centres" defined previously or is it a common phrase in the field? May need to define for clarity.
- Line 87: "dominance."
- Line 88: competitive advantage; "growth superiority" check word choice
- line 118: delete comma after present
- Line 135: "fewer than".

Discussion

- Lines 177-180: "opposed to this outlook" – awkward-restructure sentence for clarity
- Line 182: hot spots (concentrated/clustered areas) of what? More clarity needed. Maybe referring to

predicted densities?

- Line 183: "suitability of climatic conditions"

- Line 184: specify decreasing suitable areas in what regard.

Reviewer #3 (Remarks to the Author):

The ms COMMSBIO-20-3407-T by Baumbach et al describes expected shifts in major forest types in Central America due to climate change by means of SDMs. On the positive side the methods follow overall up-to-date modelling protocols (but more info on species selection is needed), and the results look robust and not overinterpreted. On the not so positive side, the results are somewhat not surprising, since it shows what one would think a priori (shrinking and fragmentation of mountaintop forests). This weakness could be anyway alleviated (see comments below). The work suffers from the common uncertainties of SDM approaches. In this sense, however I have to say that the authors write the best -most spot on and straightforward- argument I have seen in relation to SDM works, with which I could not agree more : "The use of SDMs for climate change studies is often contested due to the cascade of uncertainties [...]. While we generally agree with these concerns, we would like to stress, that the largest weakness of SDMs does not originate from wrong conceptualization, but from wrong application. Here, we limit our interpretation to exploring general patterns and put special emphasis on assessing and communicating uncertainties"

In any case I feel the authors take the "general patterns" slightly too literally, and present too coarse results. While completely agreeing that exploring just general patterns are the best way to avoid pitfalls in SDM works, I think that authors could confidently take the results slightly further. This would make the results more revealing and not trivial at all. For example it would be a true contribution to map where large scale fragmentation of forest types occurs (i.e. not only in the environmental space like in Fig2, but in the geographic). This could help to identify those areas that should be prioritized in conservation strategies (avoiding deforestation) because they are connectivity bottlenecks. In the same vein, I think the authors could put numbers or ranges to habitat loss % for each vegetation type (right now it can only be read from barplots in Fig 1).

As said, the modeling approach follows up-to-date protocols. My only important concern is the selection of species. What specific criteria was used for species selection within each forest type? General guidelines are given, but not specific criteria. But most importantly, how do the authors know that with these four species the environmental space of each forest type is adequately covered? How well each forest type is represented by the four selected species will depend on at least three factors I can think of: the niche width of each species, the niche overlap of the four species and the environmental width of the forest type. Worst case scenario it is an environmental diverse forest type represented by four species that have very narrow niches and that are very similar among them. Best case scenario is a forest type with very limited environmental heterogeneity, and represented by four species with wide niches that barely overlap among them. The authors would need to add convincing info supporting the use of the four selected species as representative of each forest type.

Point-by-point response to reviewer's comments

Modelling background

E: *“The manuscript would need to provide better context around the specific advance to understanding the impact of climate change in tropical forest ecosystems and more background information about climate and trait-based modeling in the introduction.”*

R2: *“Areas in need of improvement include adding SDM and ensemble SDM climate modeling, as well as trait-based distribution modeling background to the introduction.”*

We extended the introduction with more background information on SDMs for forecasting climatic suitability and the use of trait-based approaches in this context (lines 59-78):

“To investigate potential range shifts under different climatic states, species distribution models (SDMs) represent a commonly applied tool (Booth, 2018; Urbina-Cardona et al., 2019). State-of-the-art SDM techniques commonly build ensembles of multiple individual models to cover a broader range of algorithms and improve overall robustness of model predictions (Araujo et al., 2019). In the Central American region, SDMs have been occasionally used before (BIOMARCC-SINAC-GIZ, 2013; de Sousa 2018), yet their results are limited to single species. For better conservation planning, however, an extension of such focus studies beyond the species level is desirable. A simple way forward lies in the aggregation of many single-species models to stacked-SDMs (SSDMs) to summarize predictions across species and gain insights about more general trends in species communities (Calabrese et al., 2014; Biber et al., 2019). Nevertheless, for investigating species communities or ecosystems pure “mass stacking” SSDM approaches are impractical, since they require large computational efforts and multiply uncertainties of individual SDMs. Therefore, in contrast to such species-specific modelling, trait-based approaches have been increasingly gaining attention in ecological research. By focusing on the relationship between physiological, morphological and life-history characteristics of organisms and their environment, these approaches allow for the identification of more fundamental patterns beyond the species level (Zakharova et al., 2019). Consequently, interlinking SDMs with trait-based approaches may be particularly valuable for analyses in species-rich regions such as Central America. ”

“Species types”

R1: *“Finally, I did not agree with the use of “species types” throughout the text. In my opinion, there isn't a “type” of species if that is not based on a criteria. In this case it was “traits”. So, I strongly recommend switching throughout “species types” for “species trait-based types” or “plant functional groups.”*

We completely agree with this term being misleading. Therefore, we exchanged “species types” throughout the whole manuscript in favour of “plant functional types”(PFT) , which is a concise and scientifically established term and fits better in the context of our study.

Species selection

E: *“The manuscript would also need to provide convincing justification for the species selected to represent forest types and for the selected sample sizes.”*

R1: *“However the number of species used (Table 1) for each “species type” was quite low for the size of the area that they were simulating.”*

R3: *“My only important concern is the selection of species. What specific criteria was used for species selection within each forest type? General guidelines are given, but not specific criteria. But most importantly, how do the authors know that with these four species the environmental*

space of each forest type is adequately covered? How well each forest type is represented by the four selected species will depend on at least three factors I can think of: the niche width of each species, the niche overlap of the four species and the environmental width of the forest type. Worst case scenario it is an environmental diverse forest type represented by four species that have very narrow niches and that are very similar among them. Best case scenario is a forest type with very limited environmental heterogeneity, and represented by four species with wide niches that barely overlap among them. The authors would need to add convincing info supporting the use of the four selected species as representative of each forest type.”

We understand the reviewers' concerns. To make our species selection process more transparent and comprehensible, we added a paragraph in the Supplementary to describe this step in more detail (lines 988-1001):

*“A preliminary list of eligible species was created mainly based upon the compendium of Central American tree species by Cordero & Boshier (2003), which already featured a grouping of species into Holdridge Life Zones. We thinned this list of species “candidates” by referring to information on these species in other guides (as cited in the main text), checking for occurrence data availability (>200) in the cited data bases and retrieving and checking trait data (for wet and dry acquisitive/conservative). To avoid including highly specialized or region-specific species, we further filtered the list by widespread abundance and large overlap of species presence records with the forest types they were supposed to represent. To validate the latter two points we compared the distribution of occurrence data for each PFT with ecoregions (Olson et al. 2001), which is shown in Figure S27. Final selections were discussed and refined with the help of regional plant experts (Dr. J. Franciso Morales, Dr. Lenin Corrales and Dr. Bryan Finegan), which for example led to the decision of grouping *Weinmannia* species together due to close co-occurrence and easy confusion in the field.”*

We also added a map showing the distribution of occurrences for each PFT within ecoregions (Fig S27), which we believe adds more validity to our selection.

Our rationale for using only a small number of species for each type was based on several factors: 1) the large computational effort associated with running these kind of models, 2) the multiplication of uncertainties for each additional species and 3) our focus on traits instead of species (also see updated introduction). We highlighted these points in our updated introduction.

Occurrence data within ecoregions

Figure S27: Distribution of occurrence data within ecoregions.

Additional vegetation types

R1: “Similarly, as highlighted by the authors, some vegetation were not classified into any of the categories, and thus not considered in the result analysis. However, considering the forest dynamics of tropical environments, I found that excluding lianas and palm trees (for example) from the predictions could have biased their results. I highlight these two plant functional groups specifically due to their importance in the tropical stand composition (i.e., lianas colonize forest gaps and sometimes dominate to the point of exclusion of local trees). Thus, I wished these groups (and possibly others) were also considered in these simulated scenarios.”

Indeed our study only comprises tree PFTs, which we chose based on the most common ecosystems throughout Central America (Corrales et al. 2015). We agree that considering gap dynamics, other excluded plant forms such as lianas or also shrubs and grasses may play an important role. However, as stated in the section “Model limitations”, we do not see our models fit to simulate forest dynamics such as competition or disturbance in the first place. SDMs may only model equilibrium states in relation to different environmental conditions. For analyses that include a proper representation of competition for resources (light, nutrients, etc.), process-based models would be a more appropriate choice. We also stress this in the conclusion as an important extension and comparison to our study to shed light on climate change trends from a vegetation growth perspective:

“To extend these findings, further research on growth responses to climate change throughout different biomes and vegetation types is needed to better inform management decisions (similar to Stan et al., 2020)”

Besides these methodological limitations, we are not sure about the additional value of running more distribution models for lianas or palms. While we agree, that lianas can intensely compete with trees and suppress their regeneration (e.g. Schnitzer & Carson 2010), both usually co-exist due to the liana’s dependence on vertical structures to reach the canopy (Schnitzer & Bongers 2002). In this sense, we would argue that lianas are already indirectly included in our analyses by modelling the climatic suitability of their “host” trees. With respect to palms, these are a very diverse group of plants varying in growth form from small shrubs to canopy-forming trees. In Central America, they commonly occur interspersed in lowland and premontane rain forests, in coastal regions and less frequently in transitional areas to seasonally dry forest (Baslev et al. 2011). In our initial species list, we thus considered including palm species as representatives for the wet acquisitive or generalist types. Alas, at the time of our data search, occurrence data for Central American palm species was sparse and mostly clustered in a few regions (probably because palm seeds are mostly locally dispersed). We therefore did not include them into our species selection. Since they commonly co-occur with wet forest trees and do not form monodominant associations in Central America, we would expect suitability maps to be very similar to our projections for the wet PFTs.

Nevertheless, we acknowledge the importance of other vegetation types for the structure and composition of tropical forests and the urgent need to explore this topic further. We therefore added a paragraph to discuss their relevance (lines 230-239):

“Lastly, beyond the here modelled vegetation types, other plant groups such as lianas, palms or grasses could also alter forest structure and growth due to their role in gap dynamics after forest disturbances (Estrada-Villegas et al, 2020; Balslev et al., 2011; Ratajczak et al., 2017). For instance, recent trends of increasing liana biomass across tropical forests could be reinforced under stronger global warming, resulting in reduced tree growth and overall reduced forest carbon uptake (van der Heijden et al., 2015; da Cunha Vargas et al., 2020). Exploring the competitive relationship between these plant groups from a vegetation growth perspective under the influence of climate change and different disturbance regimes thus represents an important avenue for future research.”

Novelty of results

R1: “However, I am struggling to find the novelty of this manuscript. As written by the authors in the Discussion, their findings mirror the findings of other studies.”

We agree that our general findings don't appear surprising, since they confirm trends seen in other studies. However, we would like to argue, that there is additional value in several aspects of our study:

1. A confirmation of the results from other studies allows for more confidence and adds to the body of literature calling for attention to these alarming trends
2. A specific analysis of connectivity loss and mountaintop extinction has not been done before in the study area and is of high relevance for conservation planning (as reviewer 2 also stated)
3. The fine resolution of our results allows an interpretation at ecosystem level in a topographically complex landscape. Admittedly, this interpretation could be extended to uncover interesting small-scale effects such as fragmentation (as also suggested by reviewer 3, please see our response below).
4. Our trait-based grouping approach in combination with SDMs – although not being entirely novel – is easily transferable to other study regions for highlighting hot spots of change and only requires little data

Habitat loss metrics / Figure 2

R2: “The manuscript could also benefit from more clarity on the posited connectivity loss as connectivity loss is emphasized as a key outcome in this study. Specifically, how are the strengths of divergence measured (what metrics used) then visualized for Figure 2? The authors posit that due to shifts and divergences in predicted population densities, there would be loss of connectivity. Though I agree connectivity loss is likely, there is no analysis, example, nor background (sources) presented to support this claim. Similarly, the author’s mountaintop extinction outcome is in need of support, either through referenced sources or metrics from their study. Sources and background could be included in the discussion of implications of their study.”

R3: “While completely agreeing that exploring just general patterns are the best way to avoid pitfalls in SDM works, I think that authors could confidently take the results slightly further. This would make the results more revealing and not trivial at all. For example it would be a true contribution to map where large scale fragmentation of forest types occurs (i.e. not only in the environmental space like in Fig2, but in the geographic).”

Figure 2 represents a 2-D density plot of predicted presences, which shows their distribution over the two axes latitude and altitude. As such, no specific metrics were calculated to measure the strength of connectivity loss or mountaintop extinction. For connectivity loss we evaluated the plot visually by highlighting the subplots with arrows at the points where the density clouds split into several pieces along the latitude axis. We agree, however, that connectivity loss could be shown more clearly in the geographic space, since fragmentation may also occur along longitudes and mapping fragmentation hot spots may be of large interest for conservation planning. For this purpose we calculated habitat fragmentation for each grid cell of the study area in a 3x3 moving window based on the classification approach by Riitters et al (2000). Compared to more complex landscape metrics, this approach is easily interpretable and has been successfully used in conservation studies before (Morelli et al. 2020). The resulting fragmentation maps (Figures 3, S5 & S6) and fragmentation statistics (Table S1) highlight fragmentation hot spots and potential connectivity issues more clearly now and give improved measures. Respectively, paragraphs were added in the results (lines 154-173) and discussion (lines 204-209) to describe these outcomes.

For upslope shifts and potential mountaintop extinction, we marked in Figure 2 the upwards shift of the lower boundary of the density clouds along the altitude axis in comparison to present. Approximate upslope shifts can thus be directly read from the figure for each latitude. To give better orientation, however, we added a complementary elevation map of the region to the Supplementary (Figure S26).

Figure S26: Elevation map of the study region (data: GMTED2010). Margin plots show mean elevation as a histogram along latitudes and longitudes, respectively.

Figure 3: PFT fragmentation. Maps showing fragmentation classes for each PFT in the areas where they were predicted as dominant (CCSM4 scenarios). Dark grey marks areas, which were covered by the PFT in present-day projections, but disappeared in the scenarios (“Area loss”). For definitions of the other fragmentation classes see Methods section.

Table S1: Dominant PFT area statistics. Column 1 shows shares (%) of the study area covered by each respective PFT for the present-day projections (in italics) and changes to this number for each climate change scenario (cc=CCSM4, ha=HadGEM2-AO, mp=MPI-ESM-LR). Columns 2-6 again show how much each fragmentation class contributed to the area shares of Column 1.

PFT	scenario	Study area share [%]			Patch fraction [%]			Transitional fraction [%]			Perforated fraction [%]			Edge fraction [%]			Interior fraction [%]		
		cc	ha	mp	cc	ha	mp	cc	ha	mp	cc	ha	mp	cc	ha	mp	cc	ha	mp
Dry acq.	present	8.48			5.80			11.08			27.38			5.27			50.47		
	RCP2.6	+0.35	-1.98	+1.03	6.22	7.68	6.32	11.40	13.77	11.35	28.71	31.01	29.30	4.84	5.02	4.48	48.83	42.52	48.56
	RCP4.5	-0.29	-1.92	+0.10	6.73	6.91	6.23	12.38	13.08	11.61	31.36	31.36	30.79	4.64	4.52	4.28	44.90	44.14	47.10
	RCP8.5	-0.73	-2.01	+0.29	7.73	8.35	6.88	14.26	14.70	12.53	32.33	31.15	29.86	4.53	4.67	4.25	41.16	41.14	46.49
Dry con.	present	19.32			2.10			4.70			11.76			3.61			77.83		
	RCP2.6	+0.36	-0.30	+0.95	2.15	2.05	1.92	4.69	4.33	4.52	11.86	10.83	11.39	3.56	3.60	3.76	77.73	79.19	78.42
	RCP4.5	+0.58	-0.51	+1.55	2.06	2.06	2.01	4.48	4.31	4.36	11.13	10.44	10.62	3.49	3.48	3.59	78.83	79.71	79.41
	RCP8.5	+1.48	+0.50	+1.73	2.28	2.22	2.05	4.79	4.66	4.44	11.92	11.42	10.85	3.58	3.49	3.58	77.43	78.22	79.09
Generalist	present	5.19			14.60			19.36			32.35			7.66			26.03		
	RCP2.6	+4.12	+6.35	+5.34	10.73	8.31	10.54	16.30	14.06	15.89	28.99	27.83	28.81	8.49	8.09	8.88	35.49	41.72	35.89
	RCP4.5	+10.12	+16.95	+13.96	7.31	4.87	7.12	11.87	8.80	11.61	22.20	17.93	22.30	9.05	8.15	8.42	49.57	60.26	50.55
	RCP8.5	+20.01	+23.36	+28.98	5.02	3.74	3.09	8.73	7.13	6.16	17.00	15.12	14.83	7.40	6.74	6.34	61.85	67.26	69.58
Wet acq.	present	11.52			5.55			11.94			32.16			4.78			45.57		
	RCP2.6	+1.21	+3.39	+0.45	5.25	4.78	5.35	11.36	10.83	11.76	31.16	30.94	32.04	4.66	4.49	4.72	47.56	48.95	46.14
	RCP4.5	+2.22	+4.50	+1.91	4.40	4.88	4.70	10.21	10.67	10.65	28.94	29.53	29.21	4.37	4.19	4.40	52.07	50.74	51.04
	RCP8.5	-0.02	+1.62	-0.78	5.21	4.65	4.41	10.41	10.76	10.48	27.86	28.28	28.53	4.39	4.49	4.57	52.14	51.82	52.00
Wet con.	present	10.04			5.79			9.34			22.15			5.70			57.02		
	RCP2.6	-1.11	+0.24	-2.07	6.84	6.40	8.41	10.53	9.61	11.41	23.58	21.51	24.42	5.48	5.35	5.46	53.57	57.12	50.29
	RCP4.5	-5.13	-4.54	-4.56	11.00	10.80	11.37	14.76	14.96	15.54	27.85	26.66	28.87	5.77	5.37	5.68	40.62	42.22	38.54
	RCP8.5	-7.88	-7.31	-6.27	12.50	8.00	9.16	16.78	13.31	12.90	27.38	25.76	24.64	7.02	6.57	6.01	36.31	46.37	47.29

Table S1 continued.

PFT	scenario	Study area share [%]			Patch fraction [%]			Transitional fraction [%]			Perforated fraction [%]			Edge fraction [%]			Interior fraction [%]		
		cc	ha	mp	cc	ha	mp	cc	ha	mp	cc	ha	mp	cc	ha	mp	cc	ha	mp
Montane	present	11.28			1.91			6.44			16.56			3.40			71.69		
	RCP2.6	-1.39	-1.64	-1.53	2.07	2.05	2.03	6.55	6.62	6.75	16.39	16.71	17.15	3.31	3.41	3.56	71.69	71.20	70.50
	RCP4.5	-2.66	-3.42	-2.85	2.09	2.05	2.06	6.62	6.66	6.81	16.81	16.86	17.75	3.48	3.59	3.63	70.99	70.83	69.75
	RCP8.5	-4.49	-4.93	-4.75	2.06	2.10	2.14	7.16	7.02	7.18	18.11	18.17	18.41	3.76	3.77	3.89	68.91	68.95	68.37
Coniferous	present	2.89			11.78			21.18			34.65			6.79			25.59		
	RCP2.6	-0.25	-0.25	-0.04	11.59	11.73	11.63	21.81	21.46	21.06	34.56	34.06	34.21	6.58	6.63	6.51	25.47	26.13	26.59
	RCP4.5	-0.43	-0.56	-0.47	10.97	10.12	12.56	20.75	20.70	22.42	34.86	35.12	33.35	6.80	6.93	6.45	26.60	27.14	25.22
	RCP8.5	-0.71	-0.89	-0.70	10.83	10.79	12.24	21.03	20.38	22.02	34.50	34.52	33.65	6.70	6.68	6.61	26.94	27.63	25.48

Sample size

R2: “Adequate sample sizes are important for these types of models and n should be included in the manuscript.”

For better transparency, we added a column in Table 1 listing the number of presence records used in our study for each species. We further added a reference (Soultan & Safi 2017) to support the appropriateness of the sample sizes with regards to the chosen modelling algorithms.

Figure 1

R3: “In the same vein, I think the authors could put numbers or ranges to habitat loss % for each vegetation type (right now it can only be read from barplots in Fig 1)”.

Statistics of habitat loss have been added together with fragmentation statistics in Table S1.

References

- Corrales, L., Bouroncle, C., and Zamora, J. C. (2015). An overview of forest biomes and ecoregions of Central America. In *Climate Change Impacts on Tropical Forests in Central America*, pages 17–38. Routledge, London.
- Morelli, T. L., Smith, A. B., Mancini, A. N., Balko, E. A., Borgerson, C., Dolch, R., Farris, Z., Federman, S., Golden, C. D., Holmes, S. M., Irwin, M., Jacobs, R. L., Johnson, S., King, T., Lehman, S. M., Louis Jr, E. E., Murphy, A., Randriahaingo, H. L., Randrianarimanana, L., Ratsimbazafy, J., Razafindratsima, O. H., and Baden, A. L. (2020) The fate of Madagascar’s rainforest habitat. *Nature Climate Change*, 10:89–96. DOI: 10.1038/s41558-019-0647-x
- Riitters, K., Wickham, J., O’Neill, R., Jones, B., and Smith, E. (2000). Global-scale patterns of forest fragmentation. *Conservation Ecology* 4(2): 3. DOI: 10.5751/ES-00209-040203
- Schnitzer, S. A., and Bongers, F. (2002) The ecology of lianas and their role in forests. *Trends in Ecology & Evolution* 17(5):223:230. DOI: 10.1016/S0169-5347(02)02491-6
- Schnitzer, S. A., and Carson, W. P. (2010). Lianas suppress tree regeneration and diversity in treefall gaps. *Ecology Letters*, 13: 849-857. DOI: 10.1111/j.1461-0248.2010.01480.x
- Soultan, A., and Safi, K. (2017). The interplay of various sources of noise on reliability of species distribution models hinges on ecological specialisation. *PLoS ONE*, 12(11):e0187906. DOI: 10.1371/journal.pone.0187906.

PFT	drought tolerance	altitudinal zone	shade tolerance	wood density	leaf nitrogen	specific leaf area	seed dry mass	leaf phenology	Representative species	Sample size
dry acquisitive	▲	▼	▼	0.56 ±0.07	2.35 ±0.37	13.66 ±0.73	525.20 ±314.35	D	Brosimum alicastrum	1316
									Calycophyllum candidissimum	550
									Enterolobium cyclocarpum	905
									Pachira quinata	334
dry conservative	▲	▼	▼	0.65 ±0.06	2.69 ±0.56	11.98 ±1.10	37.00 ±20.31	SE/E	Alvaradoa amorphoides	641
									Byrsonima crassifolia	2340
									Leucaena leucocephala	1074
									Vachellia farnesiana	1125
wet acquisitive	▼	▼	▼	0.32 ±0.05	2.57 ±0.31	16.54 ±3.60	281.825 ±266.76	SE/E	Cecropia obtusifolia	686
									Ochroma pyramidale	660
									Schizolobium parahyba	348
									Vochysia ferruginea	432
wet conservative	▼	▼	▲	0.63 ±0.08	1.96 ±0.26	14.22 ±0.67	8789.65 ±6440.16	SE	Calophyllum brasiliense	760
									Carapa guianensis	314
									Dialium guianense	489
									Symphonia globulifera	599
generalist	►	▼	►	0.42 ±0.06	2.28 ±0.08	15.93 ±1.21	745.93 ±539.51	D/SE	Ceiba pentandra	810
									Guazuma ulmifolia	3225
									Simarouba amara	498
									Spondias mombin	1520
montane	▼	▲	►	0.49 ±0.03	2.08 ±0.29	8.38 ±1.31	n/a	SE/E	Alnus acuminata	1115
									Cornus disciflora	657
									Drimys granadensis	822
									Weinmannia spp.	1468
coniferous	▲	►	▼	0.50 ±0.02	n/a	n/a	28.90 ±13.19	E	Pinus ayacahuite	270
									Pinus caribaea	206
									Pinus oocarpa	618
									Pinus tecunumanii	257

Table 1: PFT composition, functional traits and sample sizes (presence records). Trait means are shown in bold with respective standard errors indicated by ±. Qualitative traits are ranked as high/intermediate/low (▲/►/▼). Leaf phenology can be deciduous (D), semi-evergreen (SE) or evergreen (E).

REVIEWERS' COMMENTS:

Reviewer #1 (Remarks to the Author):

As one of the original reviewers, I am pleased with the authors response to my comments and how they addressed them in the manuscript. I am also pleased with the additional edits based on the other reviewers and editor's feedback. I believe this paper will be a significant contribution to our current knowledge on prediction of forest dynamics based on its plant functional type composition, especially in the tropics.

Reviewer #2 (Remarks to the Author):

The authors sufficiently addressed my major concerns from previous review. I appreciate the more detailed fragmentation analyses and agree that they add credibility to the assertions of connectivity loss and mountaintop extinction. However, the fragmentation patterns in Figure 3 are difficult to see – adding additional visualizations that highlight changes across scenarios would be helpful.

Minor comments below:

Throughout: “hot spots” and “hotspots” are both used – choose one

Introduction

- lines 38-40: still awkward phrasing – restructure

Reviewer #3 (Remarks to the Author):

The authors have addressed main concerns raised in previous review.